# Behavior of Porewater Pressures in an Earth Dam by Principal Component Analysis

**Seong-Kyu Yun** [1] , **Jiseong Kim** [2], **Eun-Sang Im** [3] **and Gichun Kang** [4],*

1 Engineering Research Institute, Gyeongsang National University, Jinju 52828, Korea; tjdrb330@gnu.ac.kr
2 Department of Cadastre & Civil Engineering, Vision College of Jeonju, Jeongju 55069, Korea; kimjs@jvision.ac.kr
3 Water Energy & Infrastructure Research Center, K-Water, Daejeon 34045, Korea; esim89@kwater.or.kr
4 Department of Civil Engineering, College of Engineering, Gyeongsang National University, Jinju 52828, Korea
* Correspondence: gkang@gnu.ac.kr; Tel.: +82-55-772-1792

**Abstract:** This study deals with the utilization of the pore pressure meter for evaluating the stability of a dam through the correlation between the porewater pressure installed in the fill dam and the water level of the dam. To this end, principal components analysis was performed on a total of 18 porewater pressure meters, and the main components were classified into three groups: internal (Group A), external (Group B), and upper (Group C), on the basis of the seepage line formed within the dam body. The coefficient of correlation between the porewater pressure and water level was found to be 0.86 to 1.00, indicating a strong positive linear relationship. This means that the maintenance of the dam is possible through the pore pressure meter present in Group A. Furthermore, the regression analysis for porewater pressures and water levels resulted in a linear regression model with the coefficient of determination ($R^2$) of Group A being between 0.74 and 0.99. In particular, $R^2$ between the power water pressure installed at the base of the dam and the water level was more than 0.99. Therefore, it was shown that the prediction of the porewater pressure is possible by using the relationships with the water level, making it possible to determine the safety of the dam by comparing it with the currently measured values.

**Keywords:** fill dam; principal components analysis; porewater pressure; seepage line; regression analysis

## 1. Introduction

Among the national infrastructure facilities, dams serve important functions such as that of electricity production and flood control, as well as water supply for living, industry, and agriculture. Dams in the Republic of Korea are more than 18,000 in number, and medium-sized and small dams have been built and managed since the 1960s. Due to the construction of dams, which started from Japanese colonial era, multi-purpose dams and water supply dams that have existed for more than 30 years account for more than 60%. The aging of these dams is affecting their structural stability. In this regard, dam accidents can cause huge human/economic losses. Therefore, recently, studies on the utilization of dam measurement data have been conducted frequently for the safety (Pang et al., 2020 [1]) and maintenance of dams (K-water, 2019) [2].

According to the International Commission on Large Dams, about 150,000 cases of dam collapse and accidents have been reported worldwide, and more than 2000 since the 12th century and more than 200 after the 20th century have been reported to have caused casualties involving more than 238,000 people (ICOLD, 1995) [3]. Outside the country, more than 200 dams have collapsed in Italy, the United States, and France in the 1900s, causing more than 11,000 casualties (Jansen, 1983) [4]. In Korea, more than 100 people died in the collapse of the Hyogiri Dam in Namwon, North Jeolla Province, in 1961 (Chang et al., 1998) [5]. Yeoncheon Dam, located in Yeoncheon-gun, Gyeonggi Province, caused huge economic damage to Paju and Pocheon, as well as surrounding areas, due to dam

collapse in two instances due to overflowing caused by heavy rains in the summers of 1996 and 1998. Therefore, in order to cope with the threat of safety of the dam, the main agent managing the dam must assess the stability of the dam by installing various measuring instruments and conducting real-time or regular stability evaluations (Kang et al., 2018) [6].

In particular, the porewater pressure meter, which is installed and operated during construction of the fill dam, is an important measurement item for monitoring the barrier role of the dam along with the water seepage. In a study by the U.S. Commission on Large Dams (USCOLD, 1975) [7], 77 cases of collapse of rockfill dams in the USA were analyzed, with 44% of the damage being found to be caused by leaks and piping through the dam's body or foundation, and finding that the porewater pressure meter is an important measurement item for monitoring this type of destruction. For this reason, research on the use of porewater pressure is being actively conducted (Wang et al., 2018 [8]). The porewater pressure meter is only used in Korea to check the presence or absence of a stable barrier role after construction. The reason is that the porewater pressure meter buried inside the dam has a relatively short lifespan compared to the external instruments, since it is difficult to maintain due to aging caused by wet conditions in the meter and breaking caused by deformation of the dam due to time passing after completion. However, thanks to the recent development of measuring technology nationally and abroad, the installation and operation of instruments have been stable, and the importance of the porewater pressure meter, which is a major measurement item of the fill dam, has emerged (Kang et al., 2020) [9].

In this study, we intended to analyze the correlation between the porewater pressure installed in the fill dam and the water level of the dam, and conducted a stability evaluation to present a plan for the utilization of the porewater pressure meter for future dam safety.

## 2. Region Subject to the Study

### 2.1. Dam Subject to the Study

Figure 1 shows the overall view of Gampo Dam located in the Republic of Korea. It is a central core-type rockfill dam with a total capacity of 2.39 million m$^3$, a height of 35 m, a dam extension of 108 m, and a volume of 190,000 m$^3$. The dam was completed in 2007, and 13 years have passed since impoundment.

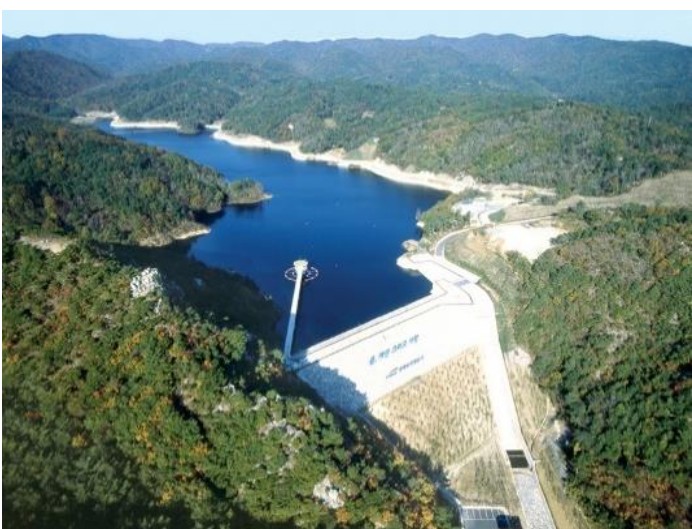

**Figure 1.** Study fill dam.

### 2.2. Installation Status of Measuring Instruments

The measuring instruments of the dam to be studied were installed for stability and behavior analysis during construction, after completion, and impoundment, and they detect stability and behavior of the dam through the analysis of measured data. Their purpose is to

be utilized for operation and maintenance of the dam for long-term use. In general, in the case of the fill dam, various instruments such as porewater pressure meter, earth pressure gauge, inclinometer, and differential settlement gauge are installed. In the case of the porewater pressure meter subject to this study, a total of 18 locations are installed in the base, core, and filter sections of the dam, as shown in Figure 2, in order to determine the appropriateness of the penetration outflow through the variation of the water pressure after impoundment. Measured data were automatically measured from 1 June 2009 after impoundment, and this study analyzed measurement data for around 10 years until 10 June 2019.

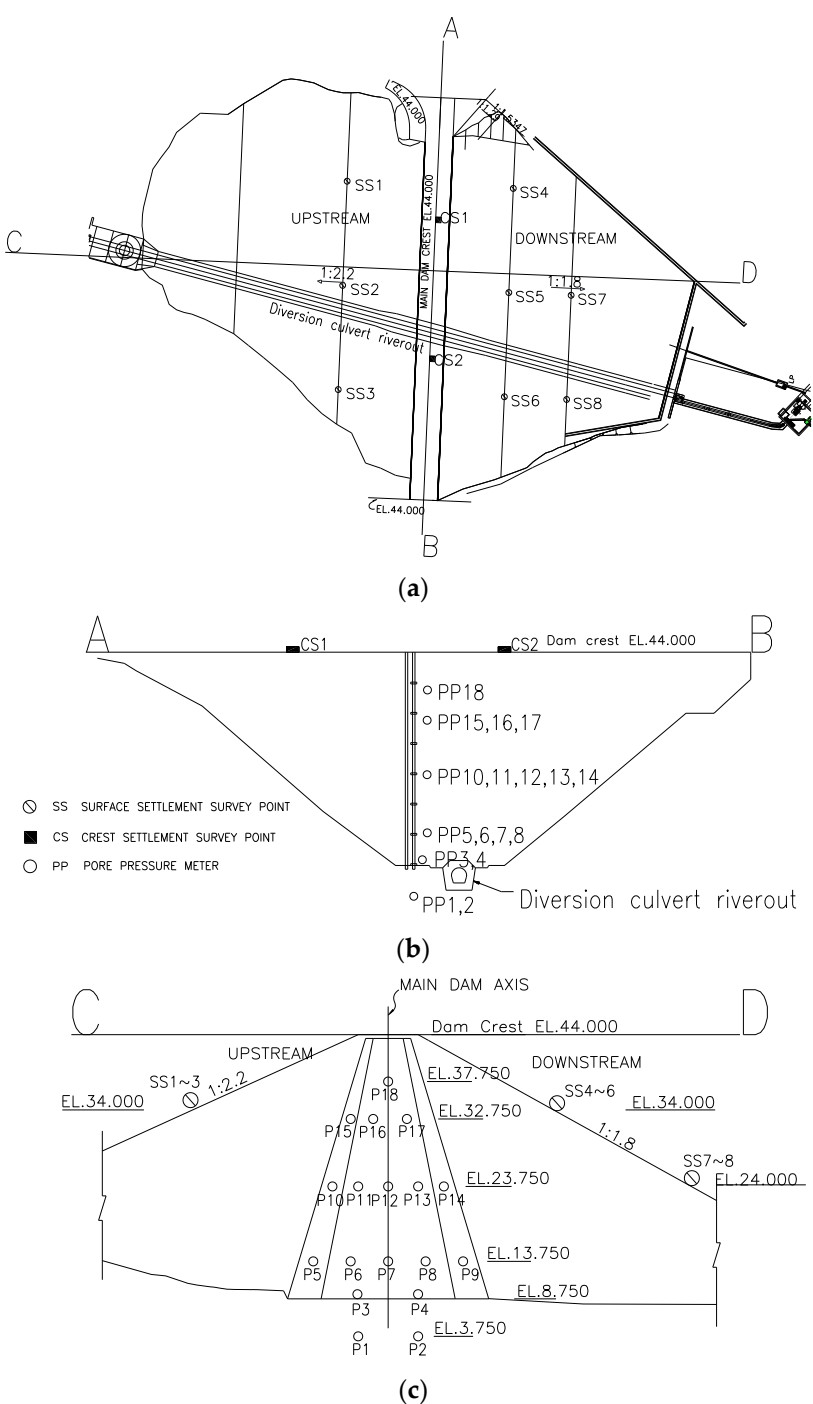

**Figure 2.** Location of porewater pressure transducers: (**a**) on the floor plan; (**b**) on the cross-section (A–B); (**c**) on the cross-section (C–D).

## 3. Measurement Status

### 3.1. Water Level

Figure 3 shows changes in water level and rainfall over time in the dam subject to the study. The missing rate of the water level during the data collection period was 0%, indicating that the data are well managed. As shown in Figure 3, it can be seen that the water level rises when rainfall increases rapidly, and that it is operated and managed within the normal high-water level (EL.40 m). The average of the water level gauge was 35.68 m, and its standard deviation was 2.97 m, while the average rainfall was 3.27 mm (std 12.13 mm), and the maximum rainfall was 234 mm.

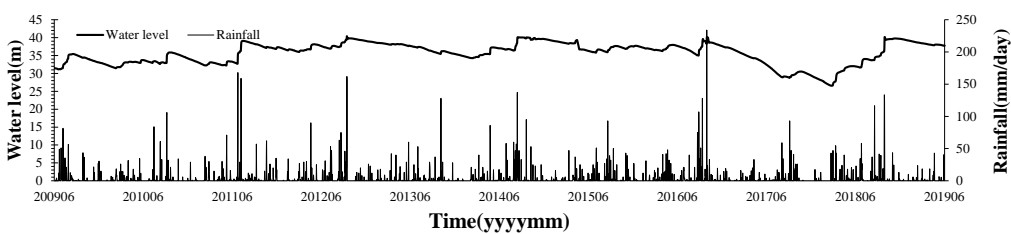

**Figure 3.** Water level and rainfall.

### 3.2. Porewater Pressure

The measurement results from 1 June 2009 to 10 June 2019 were analyzed. The average missing rate of the porewater pressure meter during the data collection period was 7.5%, and the missing frequency for each point was similar. The pore pressure meter point PP09 was excluded from the analysis as it became inoperable after completion. Figure 4 are representations of the water level and the porewater pressure at point PP01–PP18. Each analysis result can be represented by Table 1. It shows that the water pressure is high at PP01 and PP02, which are located at the bottom, affected by the seepage line formed inside the dam's body. Located in the core part of the dam, PP12 showed relatively low water pressure, which indicates that PP12 may be at the boundary of the seepage line. In the case of PP04 and PP08 installed on the downstream side of the dam's core part, it can be seen that relatively large values work in the early stages of impoundment and become smaller over time. It may be determined that the water pressure worked as the seepage line was formed during the early impoundment and the water pressure became lower as the dam became stabilized. PP13 and PP14 are determined to be located at the upper part of the seepage line, measured to have negative porewater pressure with averages of −26 kPa and −45 kPa, respectively. In the case of PP16 to PP18 installed at the top of the upper side of the dam, it was shown that the negative (−) porewater pressure mainly worked. It is installed near the water level and is considered to be the negative (−) porewater pressure caused by the unsaturated ground when the low water level becomes lower.

**Table 1.** Pore pressure gauge technical statistics analysis result.

| PP | Mean (kPa) | Standard Deviation (kPa) | Range | | |
|---|---|---|---|---|---|
| | | | Minimum (kPa) | Medium (kPa) | Maximum (kPa) |
| PP0001 | 3.33 | 0.25 | 2.53 | 3.39 | 3.73 |
| PP0002 | 2.53 | 0.22 | 1.82 | 2.58 | 2.88 |
| PP0003 | 1.38 | 0.22 | 0.78 | 1.38 | 1.93 |
| PP0004 | 0.74 | 0.25 | 0.32 | 0.67 | 1.27 |
| PP0005 | 2.11 | 0.30 | 1.18 | 2.19 | 2.55 |
| PP0006 | 1.84 | 0.25 | 0.97 | 1.90 | 2.21 |
| PP0007 | 1.39 | 0.17 | 0.72 | 1.42 | 1.64 |
| PP0008 | 0.70 | 0.23 | 0.33 | 0.64 | 1.37 |
| PP0010 | 0.89 | 0.30 | −0.09 | 0.96 | 1.36 |
| PP0011 | 0.65 | 0.27 | −0.25 | 0.71 | 1.09 |
| PP0012 | 0.32 | 0.19 | −0.34 | 0.36 | 0.58 |
| PP0013 | −0.26 | 0.07 | −0.45 | −0.27 | −0.07 |
| PP0014 | −0.46 | 0.04 | −0.53 | −0.46 | −0.36 |
| PP0015 | −0.19 | 0.23 | −0.62 | −0.15 | 0.25 |
| PP0016 | −0.40 | 0.75 | −11.22 | −0.30 | 0.27 |
| PP0017 | −0.25 | 0.28 | −4.71 | −0.22 | 0.05 |
| PP0018 | −0.30 | 0.10 | −2.45 | −0.31 | −0.13 |

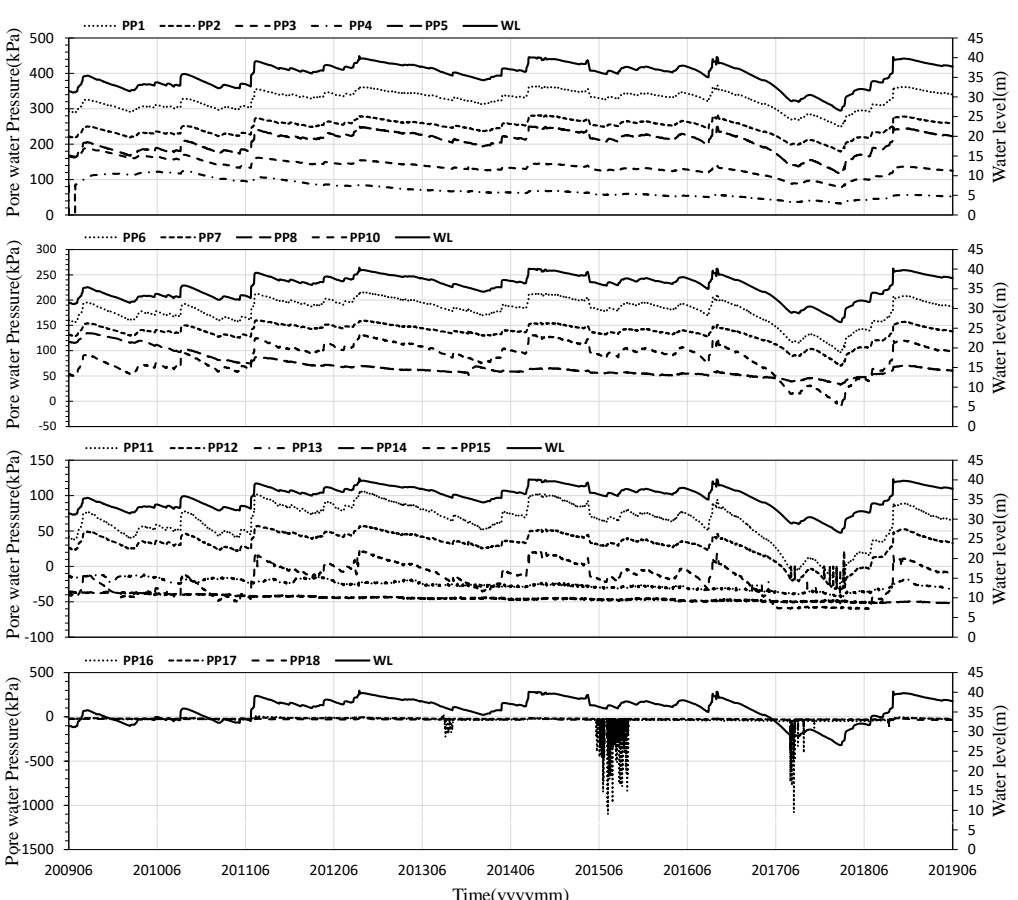

**Figure 4.** Relationship between porewater pressure and water level.

## 4. Principal Component Analysis (PCA)

Principal component analysis is one of the methods utilized in multivariate analysis. It is a method of reducing high-dimensional data to low-dimensional data. The concept was first proposed in the 1900s, similarly to the principal axis theorem theory (Person, 1901) [10], and was later established by Hotelling (1933) [11]. Unlike multidimensional scaling, which can be applied only with distance matrices between objects (Kwon, 2016) [12], principal component analysis is widely used in various disciplines such as the humanities, economics, business, and engineering (Kwon et al., 2020) [13] due to its advantages of understanding potential characteristics of variables and its enabling of spatial representation of consumer perceptions and preferences in addition to data reduction (Kim, 2016) [14].

Principal component analysis is a method of summarizing and analyzing a small number of comprehensive characteristics using linear combinations of independent variables while minimizing the loss of information. It is a technique that is often used to identify factors behind a particular idea or when it is more reasonable to deal with specific ideas and their background factors comprehensively rather than independently (Gwak and Kim, 2016 [15]; Park and Rhee, 2012 [16]). For example, it would be more reasonable and rational to identify measured items used for safety monitoring of dams by integrating them into several common factors rather than to identify them in detail by instrument.

Principal component analysis, which is a linear combination similar to regression analysis, can be expressed in the following formula (Lee, 2012 [17]; Lee and Nho, 2015 [18], Nho, 2007 [19]):

$$\begin{aligned}
z_1 &= a_{11}x_1 + a_{12}x_2 + \cdots + a_{1p}x_p \\
z_2 &= a_{21}x_1 + a_{22}x_2 + \cdots + a_{2p}x_p \\
z_n &= a_{n1}x_1 + a_{n2}x_2 + \cdots + a_{np}x_p
\end{aligned} \tag{1}$$

where $z_1 \sim z_n$, $a_{11} \sim a_{np}$, and $x_1 \sim x_p$ are the principal component, the coefficient of principal component analysis, and the independent variable, respectively.

### 4.1. Calculation of the Principal Components

The linear combination of regression analysis and principal component analysis is similar, but for regression analysis linear combinations, it is a linear combination that minimizes the independent variable linear combination deviation for the dependent variable. On the other hand, the linear combination of principal component analysis is different, as it is the linear combination minimizing the deviation by the linear combination of independent variables. Therefore, the principal component deviation is calculated as the vertical line, which is the shortest connection line to the linear combination line with the variable. The linear combination minimizing this deviation ($R$) is shown in Equation (2), which is interpreted as a linear combination minimizing the loss of information of the variables (Lee, 2012 [17]; Lee and Nho, 2015 [18], Nho, 2007 [19]).

$$R = \frac{\left| a_{11}x_1 + a_{12}x_2 + \cdots + a_{1p}x_p \right|}{\sqrt{a_{11}^2 + a_{12}^2 + a_{1p}^2}} \tag{2}$$

In order for the loss of information in principal component analysis to be minimized, the value of simultaneous equation composed by the assumption of "$a_{11} + a_{12} + \cdots + a_{1p} = 1$" is changed into the Lagrange function ($\lambda$), and the partial differential value of the squared deviation is set as zero. The formula for this is expressed in Equations (3) and (4) (Lee, 2012 [17]).

$$F = R\left(a_{11}, a_{12}, \ldots, a_{1p}\right) - \lambda\left(a_{11}, a_{12}, \ldots, a_{1p}\right) \tag{3}$$

$$\frac{\partial F}{\partial a_{11}} = 0, \ \frac{\partial F}{\partial a_{12}} = 0, \ \ldots, \frac{\partial F}{\partial a_{1p}} = 0 \tag{4}$$

In the regular equation of the principal component analysis, the information loss minimization is calculated with the Lagrange function $\lambda$ value. That is, if there are $n$

independent variables, then there are $n$ principal components that aggregate them. When the Lagrange function $\lambda$ value reaches its maximum, the information loss becomes the lowest, which is shown as the maximum of the principal component variance according to the relationship found in Equation (5) (Lee, 2012 [17]).

$$[Sum\ of\ independent\ variable\ variance] = \left[\frac{Sum\ of\ squared\ information\ loss}{measured\ number} - 1\right] + [Principal\ component\ variance] \tag{5}$$

In the principal component analysis, the primary principal component of eigenvector $(a_{11}, a_{12}, \ldots a_{1p})$ according to the maximum eigenvalue $\lambda_1$ can be expressed as $z_1 = a_{11}x_1 + a_{12}x_2 + \cdots + a_{1p}x_p$, and the proportion of this primary principal component can be expressed as Equation (6) (Lee, 2012 [17]).

$$P = \frac{\lambda_1}{\lambda_1 + \lambda_2 + \cdots + \lambda_p} \tag{6}$$

*4.2. Analysis of Principal Components and Score of Principal Components*

It is not easy to accurately interpret what the principal component means when the measured variables are abbreviated to a small number of principal components while minimizing the loss of information of the measured variables. Principal component analysis is performed according to existing studies or empirical results, and they are generally interpreted in an abstract and conceptual sense, not as measurable concepts, as the meanings are given subjectively by the research analyst on the basis of the principal component calculation coefficient of the measured variables. The analysis of the principal components is conducted on the basis of the interpretation of the coefficients of the formula comprising the principal components, and the researcher properly interprets the names and contents of the principal components in consideration of the characteristics and coefficients of the measurement variables.

If there are $n$ independent variables, there are also $n$ principal components that aggregate them, and there is no prescribed method for determining the number of principal components. Generally, the criteria for determining the number of principal components are divided into the case on the basis of the eigenvalue 1.0 of the Lagrange function, or the case where the number of components are determined at a level of 80% of cumulative proportion of the principal components cumulated from the maximum proportion of the principal components (Lee, 2012 [17]). In order highly correlated measured variables to be summarized into a small number of principal components, the principal component score for each measurement case can be calculated after calculating the coefficient of the formula and organizing the principal component calculation formula. The analysis of the principal components for the measurement case is performed according to the principal component score, and the formula for calculating the principal component score of the principal component $i$ is a functional formula according to the measurement of measured variables of the principal component coefficient and a functional formula according to the average value of measured variables of the principal component coefficient, which is expressed as Equation (7) (Lee, 2012 [17]).

$$Z(x_1, x_2, \ldots, x_p) = a_{i1}x_1 + a_{i2}x_2 + \cdots a_{ip}x_{p-} \left(a_{i1}x_1^* + a_{i2}x_2^*, \ldots, a_{ip}x_p^*\right) \tag{7}$$

where $a_{i1}, a_{i2} \ldots a_{ip}$ represents the coefficient of principal component analysis, and $x_1^*, x_2^*, \ldots, x_p^*$ represents the average value of variables.

## 5. Result of Porewater Pressure Analysis

*5.1. Linear Interpolation of Missing Data*

Figure 5 shows the raw data of PP01 and the time series data after linear interpolation. The average missing rate was 7.5%. The missing point showed a downward curve with a value of 0, as shown in the figure, with the measurement recorded as "0". As shown in

the figure, according to the time series distribution of the porewater pressure meter, it was determined that unusual behaviors such as severe vertical vibration or large amount of outliers occurring locally were not observed in the time series distribution, and therefore linear interpolation using Spline was conducted. In other words, linear interpolation was conducted for missing sections by generating Spline, connecting local peaks extracted from inflection point analysis and synthesizing the extracted raw data and the generated Spline.

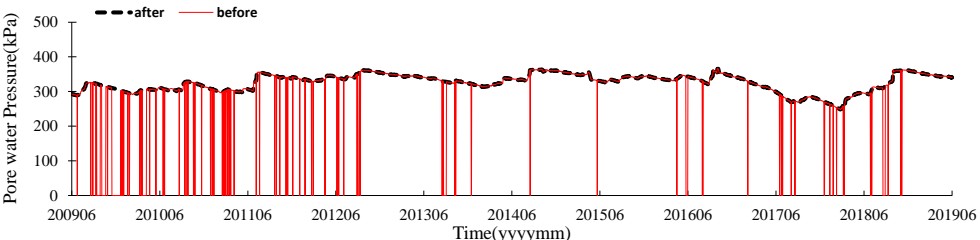

**Figure 5.** Linear interpolation for porewater pressure.

*5.2. Principal Component Analysis*

In this study, principal component analysis was performed to determine the appropriate group on the basis of the statistical similarity of the porewater pressure meters installed in the dam body under study. Although the criteria for how many principal components to be adopted have not been theoretically determined, the porewater pressure meter was summarized in three principal components by applying the generally applied grouping criteria (Lee, 2012 [17]): (1) 1 or higher correlation matrix eigenvalue (latent), (2) cumulative proportion of more than 70~80%. Figure 6 represents the component chart according to the three selected principal components (Table 2). The results of the group classification of the porewater pressure meter performed from the component score coefficient matrix results are shown in Table 3. According to the component scores in Table 3, PP01, PP02, PP03, PP05, PP06, PP07, PP10, PP11, PP12, and PP15 were classified as Group A; PP04, PP08, PP13, and PP14 as Group B; and PP16, PP17, and PP18 as Group C.

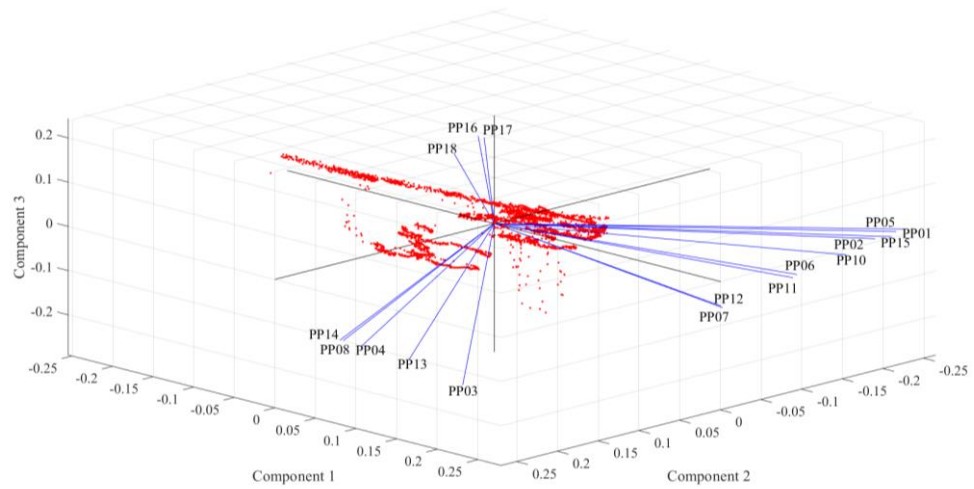

**Figure 6.** Results of the PCA.

**Table 2.** Latent and cumulative scores for PCA.

| Component | Latent | Cumulative (%) |
| --- | --- | --- |
| Comp.1 | 9.79 | 60 |
| Comp.2 | 4.23 | 80 |
| Comp.3 | 2.10 | 90 |

**Table 3.** Coefficient matrix for component scores and group distributions for porewater pressures.

|  | Comp.1 | Comp.2 | Comp.3 | Group |
|---|---|---|---|---|
| PP0001 | 0.28 | −0.23 | 0.01 | A |
| PP0002 | 0.29 | −0.22 | 0.00 | A |
| PP0003 | 0.25 | 0.29 | −0.14 | A |
| PP0004 | 0.16 | 0.39 | −0.14 | B |
| PP0005 | 0.28 | −0.22 | 0.01 | A |
| PP0006 | 0.31 | −0.10 | −0.03 | A |
| PP0007 | 0.32 | 0.00 | −0.06 | A |
| PP0008 | 0.13 | 0.41 | −0.15 | B |
| PP0010 | 0.30 | −0.17 | −0.01 | A |
| PP0011 | 0.31 | −0.10 | −0.04 | A |
| PP0012 | 0.32 | 0.00 | −0.07 | A |
| PP0013 | 0.20 | 0.33 | −0.13 | B |
| PP0014 | 0.12 | 0.41 | −0.15 | B |
| PP0015 | 0.28 | −0.19 | 0.01 | A |
| PP0016 | 0.09 | 0.14 | 0.59 | C |
| PP0017 | 0.11 | 0.14 | 0.59 | C |
| PP0018 | 0.12 | 0.21 | 0.45 | C |

Figure 7 shows the location of three group-specific porewater pressure meters classified on the basis of principal component analysis. As shown in the figure, the porewater pressure meters classified as Group A were located primarily in the upper/lower part of the upstream dam, whereas those classified as Group B were mainly distributed in the downstream part/intermediate part of the dam, and those classified as group C were mainly distributed in the upper part of the upstream dam. They were classified as internal, external, and upper on the basis of the seepage line formed within the dam body.

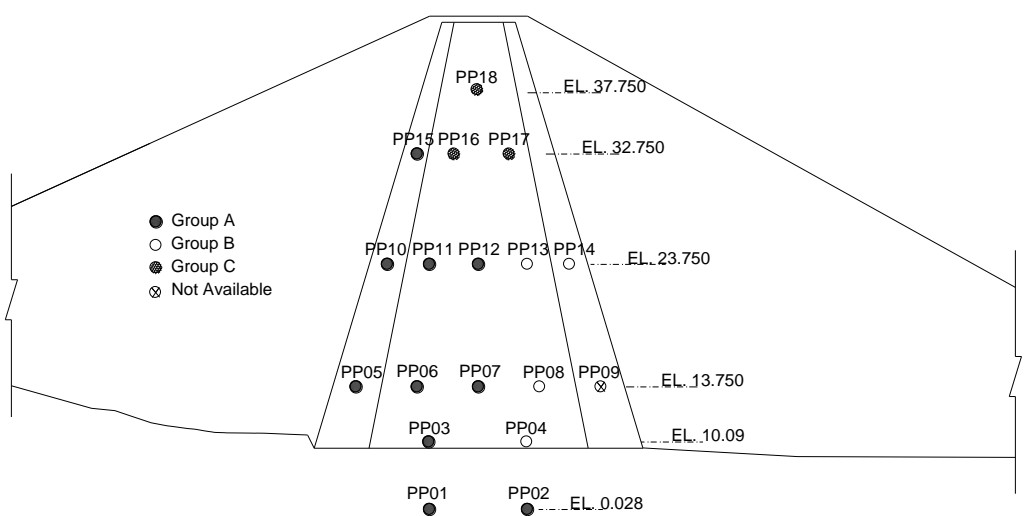

**Figure 7.** Distributions of porewater pressure transducers by PCA.

*5.3. Correlation Analysis by Group*

As for the correlation analysis for the porewater pressure meter, the instrument-specific correlation analysis within the group and the correlation analysis between the classified group and water level were conducted. Correlation analysis in Group A (PP01, PP02, PP03, PP05, PP06, PP07, PP10, PP11, PP12, PP15), Group B (PP04, PP08, PP13, PP14), and Group C (PP16, PP17, PP18) is shown in Table 4. The correlation coefficient in Group A showed a strong positive correlation, except for PP03. From these results, we believe that individual instruments in Group A may have been complementary to each other when they missed the data or exhibited mechanical abnormal behavior. For Group B and Group C, the correlation coefficients between the instruments in the group were 0.83 to 0.92 and

0.67 to 0.90, respectively, indicating relatively high positive correlations but no correlations with other groups. Furthermore, the relationship between the porewater pressure meter and the water level can be seen in Group A, with a correlation coefficient of 0.86 to 1.00, except for PP03, indicating a strong positive linear relationship with the water level. In addition, Group A can be classified into three groups according to the size of the correlation coefficient on the basis of the results of correlation analysis with water level. Group A-a (PP01, PP02, PP05) showed the highest correlation with the correlation coefficient, with a water level of 0.99, while Group A-b (PP06, PP10, PP11, PP15) had the correlation of 0.94–0.96, and Group A-c (PP07, PP12) had the correlation of 0.86–0.87. In other words, it was shown to have a high correlation with water level and to perform similar behaviors. On the other hand, Group B and Group C showed low correlations with water levels of −0.09 to 0.22 and 0.14 to 0.18, respectively, indicating that they did not correlate with water levels. Therefore, it is believed that the risk factors of dam safety such as increased penetration pressure due to the creation of flow paths inside the dam's body could be detected in advance from the porewater pressure meter in Group A, which showed no change in water level and significant change in porewater pressure.

**Table 4.** Relationships between water level and porewater pressures in the groups.

| Group | | Porewater Pressure Gauge, PP | Correlation Coefficient, r | |
|---|---|---|---|---|
| | | | Correlation Coefficient within Group | Correlation Coefficient between Group and Water Level |
| A | a | PP01, PP02, PP05 | >0.99 | >0.99 |
| | b | PP06, PP10, PP11, PP15 | 0.94~0.99 | 0.94~0.96 |
| | c | PP07, PP12 | 0.99 | 0.86~0.87 |
| | * | PP03, | 0.40~0.78 | 0.38 |
| B | | PP04, PP08, PP13, PP14 | 0.83~0.92 | −0.09~0.22 |
| C | | PP16, PP17, PP18 | 0.67~0.90 | 0.14~0.18 |

\* It couldn't be classified as any group because reliability of the instrument was degraded.

Figure 8 shows a comparison of the measured porewater pressure and water level for each group. As shown in Figure 8a,c,e, the size of the porewater pressure in Group A varied depending on the location of the installation, but it tended to be similar to the water level. In other words, the trend of water level was consistent with the porewater pressure, represented in the z-score distribution chart of Group A shown in Figure 8b,d,f. This indicates that Group A had a very high correlation with water level as well as with the correlation between instruments in the group, as mentioned above. On the other hand, as shown in Figures 9 and 10, Group B and Group C did not show much of a correlation with water levels. Figure 11 shows changes in the seepage line of the dam prepared from the measurement results of the porewater pressure installed in the dam following the changes in water level. As shown in Figure 11, Group A-a existed inside the line, and Group A-b was included inside the line when the water level was 37.75 m. If the water level was relatively low at EL. 26.55 m, PP11 and PP12 would be outside the boundary of the seepage line, and therefore the porewater pressure would not normally be measured, whereas PP11 and PP12 would be included inside the seepage line in a normal high water level of EL.40 m. In other words, it is believed that the porewater pressure is related to the path of the seepage line caused by changes in water level.

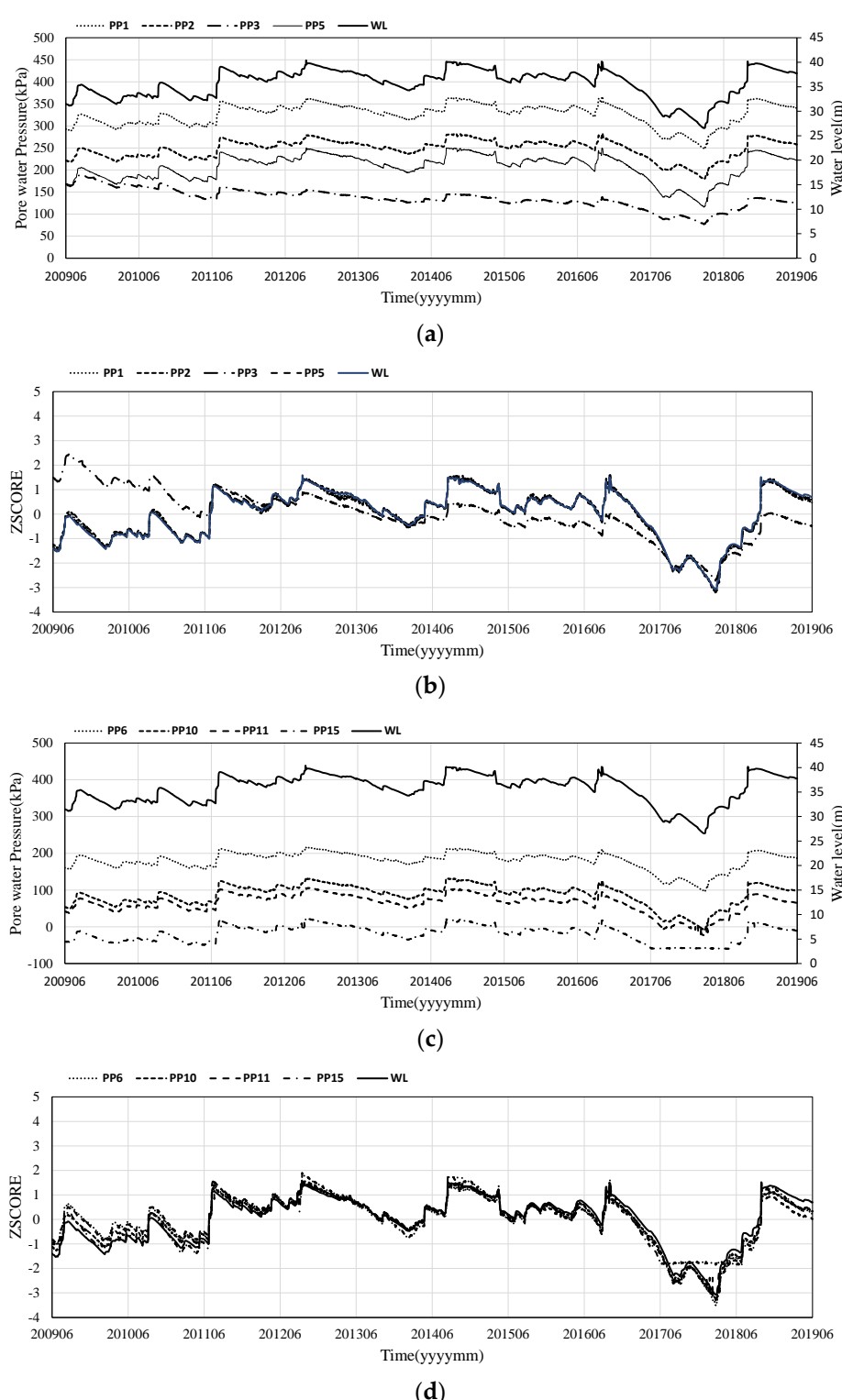

**Figure 8.** *Cont.*

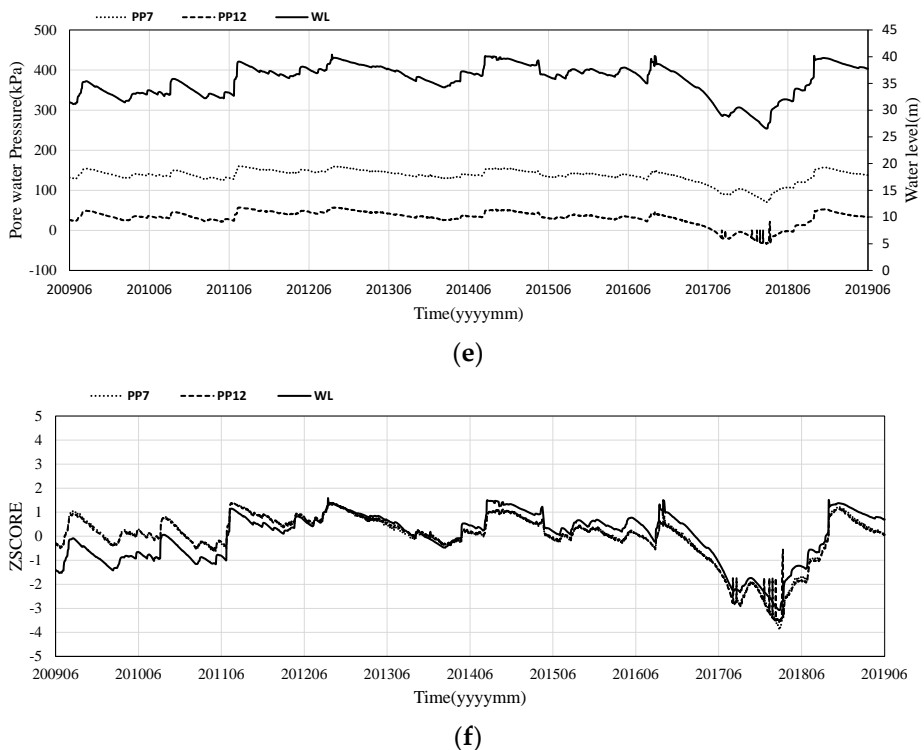

**Figure 8.** Relationships between water level and porewater pressure for group A: (**a**) comparisons with water level and Group A-a; (**b**) z-score for group A-a with water level; (**c**) comparisons with water level and Group A-b; (**d**) z-score for group A-b with water level; (**e**) comparisons with water level and Group A-c; (**f**) z-score for group A-c with water level.

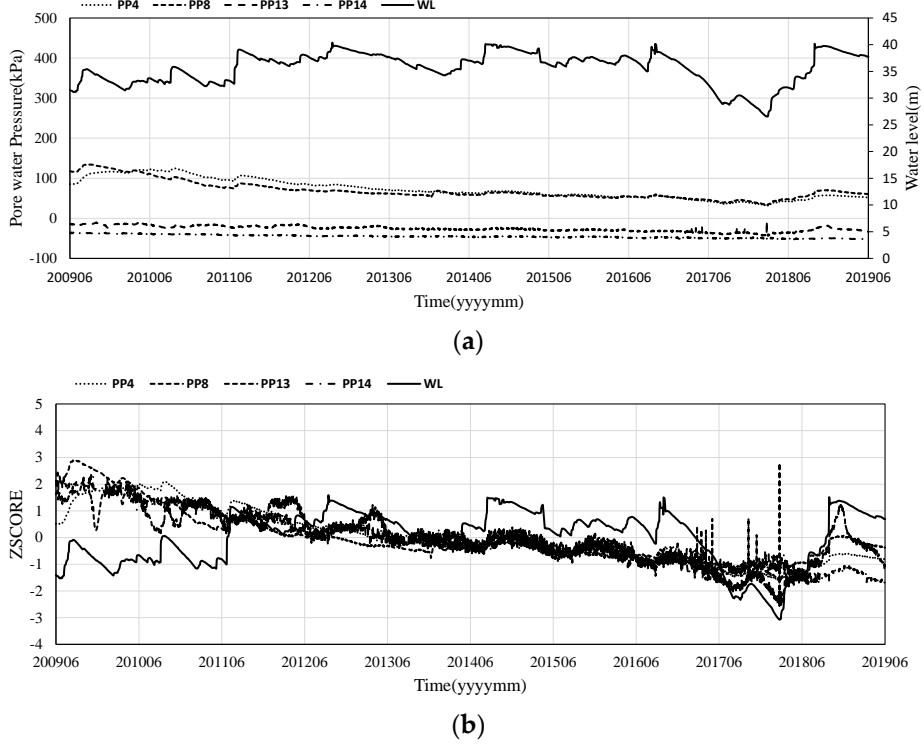

**Figure 9.** Relationships between water level and porewater pressure for group B: (**a**) comparisons with water level and Group B; (**b**) z-score for Group B with water level.

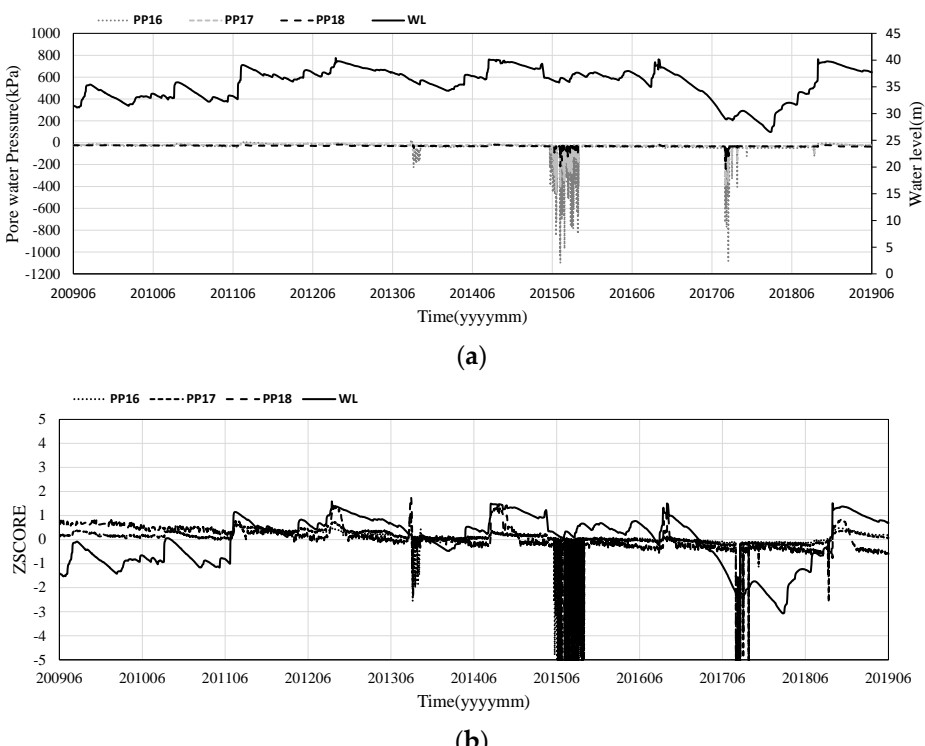

**Figure 10.** Relationships between water level and porewater pressure for Group C: (**a**) comparisons with water level and Group C; (**b**) z-score for Group C with water level.

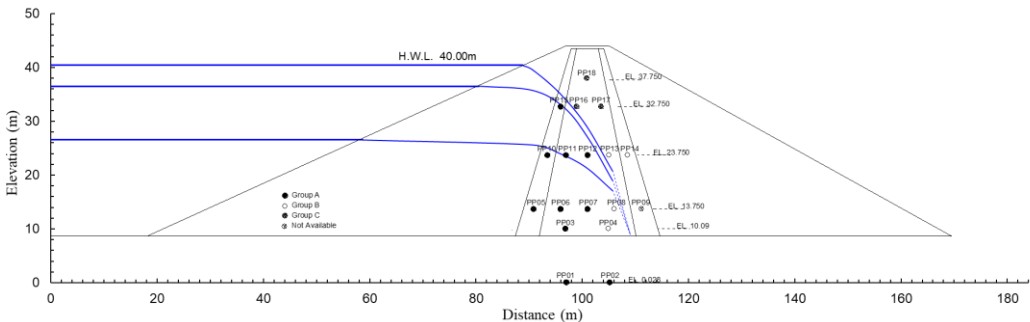

**Figure 11.** Expected phreatic line drawn from measured porewater pressures.

For PP03, the porewater pressure over time tended to be similar to water level, as shown in Figure 8a. However, as shown in Figure 8b, the z-score of PP03 was relatively large at the beginning after its completion and relatively small after about 2014, and therefore it was determined that the reliability of the instrument was degraded. That is, it was classified as Group A by the principal component analysis, but as shown in Table 4, the correlation coefficients between the instruments and the water levels in the group were 0.40~0.78 and 0.38, respectively, and was excluded from the regression analysis of following section.

## 5.4. Regression Analysis

Since the porewater pressure in Group A showed high coefficient of correlation with water level, we propose a model that can predict the porewater pressure according to the changes in water level by conducting a regression analysis on water level and porewater pressure meter. The regression analysis was performed for the case where the water level was set as an explanatory variable, and each porewater pressure was set as a response variable. Table 5 summarizes the results of the development of the primary linear regression model with the porewater pressure meter and water level. Figure 12 illustrates the results

of the development of the primary linear regression model, showing only the results for the highly correlated Group A. As shown in Figure 12a,b,d, PP01, PP02, and PP05 were classified as Group A-a with an $R^2$ of 0.99 or more, drawing on highly related relation formula. The $R^2$ of Group A-b was 0.8927 to 0.9676, indicating relatively high correlation. On the other hand, the $R^2$ of Group A-c was 0.7495 to 0.7429, indicating a relatively low correlation with the water level among Group A, but the group showed a stable distribution of each measured value around the developed regression line. For Groups B and C, the $R^2$ was distributed from 0.001 to 0.0497, and 0.0187 to 0.0331, respectively, indicating no correlation with water level. As summarized in Table 5, the determination coefficients of the primary linear regression model with the porewater pressure meter and water level were analyzed to satisfy the appropriate explanatory power for Group A only, and the porewater pressure was determined to be predictable with changes in the water level. In other words, it is possible to predict porewater pressure when the water level is fixed as an explanatory variable and to determine the safety of the dam by comparing it with the currently measured value. In addition, an appropriateness test was conducted through an F-test that was performed for each regression model developed, which was also analyzed to satisfy statistical goodness of fit in Group A, aside from Group B and Group C. In addition, the variability of the measured value tended to increase as the installation position of the porewater pressure meter became higher due to the effect of the water level. It is determined that changes in the seepage line of the dam body due to the fluctuation of the water level have a significant impact on the measured value.

**Table 5.** Regression analysis between water level and porewater pressures.

| y | x | $y = \alpha + \beta x$ | | $R^2$ | F-Test | | |
|---|---|---|---|---|---|---|---|
| | | $\alpha$ | $\beta$ | | F | *p*-Value | Test |
| PP0001 | | 30.37 | 8.297 | 0.996 | $9.8 \times 10^5$ | 0.000 | ok |
| PP0002 | | −6.3588 | 7.1313 | 0.995 | $3.75 \times 10^5$ | 0.000 | ok |
| PP0003 | | 35.734 | 2.8063 | 0.146 | - | - | NG |
| PP0005 | | −140.71 | 9.7398 | 0.997 | $1.06 \times 10^6$ | 0.000 | ok |
| PP0006 | WL | −97.607 | 7.7965 | 0.911 | $3.76 \times 10^4$ | 0.000 | ok |
| PP0007 | | −41.599 | 4.9886 | 0.749 | $1.09 \times 10^4$ | 0.000 | ok |
| PP0010 | | −261.76 | 9.7793 | 0.968 | $1.09 \times 10^5$ | 0.000 | ok |
| PP0011 | | −240.91 | 8.544 | 0.893 | $3.04 \times 10^4$ | 0.000 | ok |
| PP0012 | | −157.17 | 5.2975 | 0.743 | $1.06 \times 10^4$ | 0.000 | ok |
| PP0015 | | −276.24 | 7.2211 | 0.918 | $4.08 \times 10^4$ | 0.000 | ok |

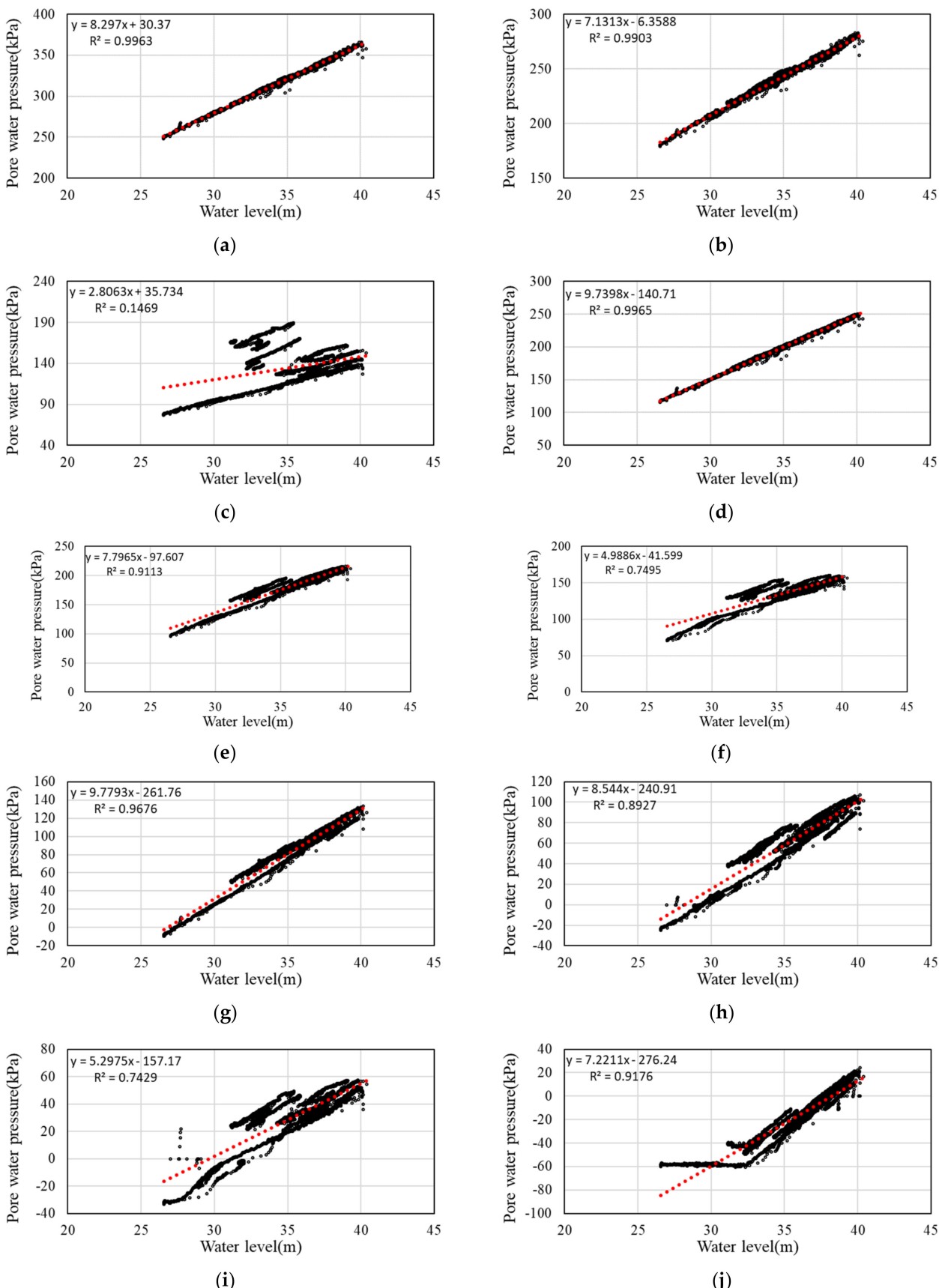

**Figure 12.** Regression analysis between water level and porewater pressures of Group A. (**a**) PP01; (**b**) PP02; (**c**) PP03; (**d**) PP05; (**e**) PP06; (**f**) PP07; (**g**) PP10; (**h**) PP11; (**i**) PP12 and (**j**) PP15.

## 6. Conclusions

In this study, the following results were obtained through analysis of the correlation with the dam water level for the porewater pressure meter utilized to predict leakage and piping of the fill dam.

1. As the result of linear interpolation for missing porewater pressure and principal component analysis, we determined the three groups: middle/lower part of upstream dam (Group A), middle part of downstream dam (Group B), and upper part of the upstream dam (Group C). Similar behaviors were shown between porewater pressure meters within a group.

2. The correlation analysis within Group A present inside the seepage line showed a correlation of 0.94 or higher, which is considered to be complementary.

3. The primary linear regression analysis of Group A, satisfying a significant correlation between water level and porewater pressure, showed the determination coefficient ($R^2$) in the range of 0.75 to 0.99, satisfying high explanatory power, and statistical goodness of fit was also found to be significant at the significance level of 5%.

4. Through the regression analysis, we found the $R^2$ of Group A-a and Group A-b to be 0.99 or more at 0.8927 and 0.9676, respectively, showing a relatively high correlation. The proposed regression analysis can predict the porewater pressure and the seepage line at constant water level.

5. Therefore, it is possible to predict the porewater pressure when there is a change in the porewater pressure at a constant dam water level, as well as when the low water level is fixed as an explanatory variable and pre-detection of the threat to dam safety due to leakage or piping inside the body is possible. Thus, the comparison with currently measured values can determine the safety of the dam.

From the results of this study, we expect that maintenance by the porewater pressure meter in Group A is required and that an early warning system can be operated using the model formula derived above for real-time safety monitoring of dam safety.

**Author Contributions:** Conceptualization, methodology, G.K. and S.-K.Y.; modeling tests and data curation, J.K. and E.-S.I.; writing—original draft preparation, S.-K.Y. and G.K.; writing—review and editing, G.K. and S.-K.Y.; supervision, G.K.; funding acquisition, G.K. All authors have read and agreed to the published version of the manuscript.

**Funding:** This research was supported by the Basic Science Research Program through the National Research Foundation of Korea (NRF) funded by the Ministry of Education (no. 2020R1I1A3067248).

**Institutional Review Board Statement:** Not applicable.

**Informed Consent Statement:** Not applicable.

**Data Availability Statement:** The data presented in this study are available on request from the corresponding author. The data are not publicly available due to privacy reasons.

**Conflicts of Interest:** The authors declare no conflict of interest.

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
