# Peer review of "Behavior of Porewater Pressures in an Earth Dam by Principal Component Analysis"

_water, doi:10.3390/w14040672_

Round 1
Reviewer 1 Report
According to the long-term observation data of pore water pressure meter, this paper makes principal component analysis, and determines the correlation between pore water pressure and dam water level through regression analysis, and gives some interesting and meaningful conclusions based on the above research results.However, there are issues that should be addressed before the paper could be recommended for publication:
1.In line 123, it should be PP16, not PP016.
2.Please explain Formula(1) in more detail. What do the subscripts n and p represent?
3.Formula(5) is not like a scientific formula. Please normalize it with mathematical symbols.
4.In line 233, one of the grouping criteria is "cumulative proportion of more than 70~80%", but Cumulative in Table2 is less 1%, please explain it.
5. The English sentence pattern of this paper still needs to be improved, and the use of some words and punctuation marks needs to be considered, especially in part of abstract.
Reviewer 2 Report
The article describes an analytical tool to connect measurements of pore water pressure in dams, together with water level, with the dam stability. The works presents an interesting application of an effective mathematical/statistical methodology to a clear practical problem which acquires even more importance as it deals with improving the safety of hydroelectric energy production, an important application in the current trend towards energy production decarbonisation.
The authors clearly explain the methodology employed and a tentative explanation of the meaning of the principal components grouping is offered.
The manuscript is worthy of publication in Water, after taking care of a few minor revisions, detailed hereafter.
- section 2.1 line 73: the dam is quantitatively described but not named, nor its location given. Since it does not appear to be a matter of confidentiality (a photo is given) this seems an oversight, while if it indeed it is, the reason for it shold clearly be stated.
- Figure 2: the figure print quality is poor, and the digital visualization is also not great. If the authors have time, could they produce a better version? Trivially, even just thicker lines and bigger numbers would suffice.
Also a clearer diagram would help, with more detailed representation of the embankment/water/dam, especially with respect to the position of sensors PP1,2 - where are those placed, precisely?
While to a reader used to studying dams this could be clear at a first glance, it is still worth the effort to make diagrams and figures in published research understandable and clearly explained as wide an audience as possible.
Maybe it would help to merge figures 2 and 4 in a same figure, with the two views side-by-side, to give a more complete overview.
- Figure 3: the axis labels are inverted, right? If max rainfall is ~230 mm and water level avg. is ~36m, I think the captions should be inverted (rainfall to the right and water lvl. to the left)
Round 2
Reviewer 1 Report
The manuscript has been modified correctly, and the modified part meets the requirements. However, the expression of porewater pressure and dam mechanical characteristics in this paper is still not perfect, and the references cited in the manuscript are not rich enough. The following references will improve this manuscript. Please refer to and quote these references appropriately.
[1] Seepage and slope stability modelling of rainfall-induced slope failures in topographic hollows. Geomatics, Natural Hazards and Risk, 2014.
[2] Stability Analysis of Partially Submerged Landslide with the Consideration of the Relationship between Porewater Pressure and Seepage Force. Geofluids, 2018, 2018:1-9.
[3] Seismic time-history response and system reliability analysis of slopes considering uncertainty of multi-parameters and earthquake excitations. Computers and Geotechnics, 2021, 136(2):104245.
[4] Fragility analysis of high CFRDs subjected to mainshock-aftershock sequences based on plastic failure - ScienceDirect. Engineering Structures, 206.
[5] Bol, Ertan. The influence of pore pressure gradients in soil classification during piezocone penetration test[J]. Engineering Geology, 2013, 157:69-78.
Author Response
In consideration of the content, only 2 out of 5 references were added.